# Cracking the Floral Quartet Code: How Do Multimers of MIKC^C^-Type MADS-Domain Transcription Factors Recognize Their Target Genes?

**DOI:** 10.3390/ijms24098253

**Published:** 2023-05-04

**Authors:** Sandra Käppel, Florian Rümpler, Günter Theißen

**Affiliations:** Matthias Schleiden Institute/Genetics, Friedrich Schiller University Jena, 07743 Jena, Germany; sandra.kaeppel@uni-jena.de (S.K.); florian.ruempler@uni-jena.de (F.R.)

**Keywords:** base readout, floral quartet-like complex, keratin-like domain, MADS domain, MIKC-type protein, shape readout, transcription factor

## Abstract

MADS-domain transcription factors (MTFs) are involved in the control of many important processes in eukaryotes. They are defined by the presence of a unique and highly conserved DNA-binding domain, the MADS domain. MTFs bind to double-stranded DNA as dimers and recognize specific sequences termed CArG boxes (such as 5′-CC(A/T)_6_GG-3′) and similar sequences that occur hundreds of thousands of times in a typical flowering plant genome. The number of MTF-encoding genes increased by around two orders of magnitude during land plant evolution, resulting in roughly 100 genes in flowering plant genomes. This raises the question as to how dozens of different but highly similar MTFs accurately recognize the *cis*-regulatory elements of diverse target genes when the core binding sequence (CArG box) occurs at such a high frequency. Besides the usual processes, such as the base and shape readout of individual DNA sequences by dimers of MTFs, an important sublineage of MTFs in plants, termed MIKC^C^-type MTFs (M^C^-MTFs), has evolved an additional mechanism to increase the accurate recognition of target genes: the formation of heterotetramers of closely related proteins that bind to two CArG boxes on the same DNA strand involving DNA looping. M^C^-MTFs control important developmental processes in flowering plants, ranging from root and shoot to flower, fruit and seed development. The way in which M^C^-MTFs bind to DNA and select their target genes is hence not only of high biological interest, but also of great agronomic and economic importance. In this article, we review the interplay of the different mechanisms of target gene recognition, from the ordinary (base readout) via the extravagant (shape readout) to the idiosyncratic (recognition of the distance and orientation of two CArG boxes by heterotetramers of M^C^-MTFs). A special focus of our review is on the structural prerequisites of M^C^-MTFs that enable the specific recognition of target genes.

## 1. MADS-Domain Transcription Factors—A Primer

MADS-domain proteins represent a eukaryote-specific family of transcription factors [1]. These MADS-domain transcription factors (MTFs) play important roles in the development and physiology of plants, animals and fungi, and possibly in almost all other eukaryotes, comprising diverse groups such as ciliates, trypanosomes, radiolarians and many more [2,3,4]. As for all transcription factors, their mode of DNA binding is a crucial aspect of the mechanism by which they recognize target genes and is hence of great biological interest.

The defining feature of all MTFs is the presence of a highly conserved DNA-binding domain, the MADS domain (Figure 1) [2]. The MADS domain has a length of approximately 60 amino acids and, accordingly, is encoded by an approximately 180-nucleotide-long DNA sequence termed the MADS box. MTFs are encoded by MADS-box genes, which were named based on four ‘founding family members’: *MINICHROMOSOME MAINTENANCE 1* (*MCM1*) from *Saccharomyces cerevisiae* (brewer’s or baker’s yeast), *AGAMOUS* (*AG*) from *Arabidopsis thaliana* (*A. thaliana*; thale cress), *DEFICIENS* (*DEF*) from *Antirrhinum majus* (*A. majus*; snapdragon) and *SERUM RESPONSE FACTOR* (*SRF*) from *Homo sapiens* (human) [5].

The DNA-binding MADS domain folds into a characteristic, highly conserved structure involving an N-terminal random coil (N-extension) and a long α-helix as the DNA contacting layer that makes DNA contacts in the minor and major grooves, respectively, and two β-strands connected by a β-turn (Figure 1) [8,9,10]. MTFs bind to DNA with a strength and sequence specificity sufficient for their biological function only as homo- or heterodimers, not as individual proteins [8,9].

The recognition of the DNA of target genes by MADS-domain proteins is a remarkable process involving diverse types of protein–DNA and protein–protein interactions (Figure 2) [11,12]. However, at the core of the typical recognition site of all MADS-domain protein dimers is a 10-bp-long DNA element termed the CArG box (for C-A-rich-G). Based on the study of the DNA-binding specificity of the human MADS-domain protein SERUM RESPONSE FACTOR (SRF) [13], the so-called ‘SRF-type CArG-box motif’ was defined as 5′-CC(A/T)_6_GG-3′, which could be considered the canonical CArG box. One-base-pair (bp) deviations are usually tolerated in many binding events, and some MTFs even prefer binding to a more deviating sequence, 5′-C(A/T)_8_G-3′, termed the ‘N10-type’ or ‘MEF2-type CArG box’ [14,15].

How did extant MADS-box genes and MTFs, with their unique fold of the DNA-binding MADS domain and recognition of a specific *cis*-regulatory element (CArG box), originate?

## 2. A Very Brief History of MADS-Box Genes

MADS-box genes seem to exist only in eukaryotes, and there is evidence that the MADS box originated from a DNA sequence encoding a region of subunit A of topoisomerase IIA in the stem group of extant eukaryotes [1]. Even before the diversification of crown group eukaryotes had started, a gene duplication led to two conserved lineages of MADS-box genes, termed Type I and Type II genes, so that, with some exceptions, the genomes of almost all eukaryotes have now both Type I and Type II MADS-box genes [16,17]. The Type I and Type II genes of animals and humans are arguably better known as the *SERUM RESPONSE FACTOR*-like (*SRF*-like) and *MYOCYTE ENHANCER FACTOR 2*-like (*MEF2*-like) genes, respectively.

For all land plant (embryophyte) genomes that have been investigated so far, both Type I and Type II genes have been annotated. Whether the Type I genes of plants are truly homologous to those of animals and fungi beyond the fact that they are MADS-box genes is questionable, however [18,19]. In any case, flowering plant Type I genes have experienced faster birth-and-death evolution than Type II MADS-box genes in angiosperms [20] and other plants [21]. They deviate in their evolutionary dynamics also from both animal and fungal Type I genes, in that they originated and were lost more rapidly than the other, highly conserved gene types [21]. In contrast, a relatively close relationship between the *MEF2*-like genes of animals and plant Type II genes is quite well supported [16].

For quite a while, all Type II MTFs of plants that had been identified possessed a characteristic and unique structure comprising four domains, the DNA-binding MADS domain (M), the intervening I domain (I), the keratin-like domain (K) and the variable C-terminal domain (C) [19,22,23]. These MTFs were hence termed MIKC-type proteins, and the corresponding genes MIKC-type genes. However, the analysis of the genomes of diverse charophytes—a grade of green algae that represents the closest relatives of land plants (embryophytes)—revealed that they contain Type II gene lineages that have K boxes and others that do not [24,25,26], implying that the terms ‘Type II genes of plants’ and ‘MIKC-type genes’ should not be used synonymously anymore. This finding also corroborates the view that the absence of a K box is not a sufficient criterion for classifying a plant MADS-box gene as Type I (as, unfortunately, can sometimes be seen in the literature). These insights are important in light of the fact that the K domain provides some unique features to MTFs.

The K domain folds into coiled-coil domains involved in dimeric and tetrameric protein–protein interactions (Figure 1) [7,11]. These interactions underlie the versatile combinations of some MIKC-type proteins that are required for combinatorial functions and target site recognition. The capacity to combine and to constitute ‘floral quartet-like complexes’ (FQCs) composed of four MIKC-type proteins binding as a tetramer to two sites (typically CArG boxes) on target gene DNA is of functional importance in planta, e.g., for the establishment of floral determinacy in the angiosperm *Arabidopsis thaliana* [27]. FQC formation may have contributed to the fact that some MIKC-type MTFs were involved in the control of many developmental processes in flowering plants. These processes include developmental phase changes and the control of organ identity, as reviewed by [11,17,28].

MIKC-type genes have so far been found in all five major clades of charophytes, but not in chlorophytes yet, corroborating the view that they are a genuine ‘synapomorphy’ (shared derived trait) of streptophytes (i.e., charophytes + embryophytes) [24,25,26,29]. This finding strongly suggests that MIKC-type genes originated in the stem group of extant streptophytes when a Type II MADS-box gene acquired a K box by unresolved mutation or recombination events. Whole-genome analyses revealed that there are very few MIKC-type genes in extant charophyte species [25,26,29], suggesting that there possibly might have been only one MIKC-type gene in the most recent common ancestor (MRCA) of extant land plants.

The ancestral MIKC-type gene present in charophytes was duplicated in the stem group of extant embryophytes (land plants), resulting in the lineages of MIKC^C^-type and MIKC*-type genes [17,29,30,31].

The number of MIKC-type genes increased strongly during land plant evolution, typically by the preferential retention and diversification of genes after whole-genome duplications [17,32]. For example, there are only two MIKC-type genes in the liverwort *Marchantia polymorpha*, and 17 in the moss *Physcomitrium patens*, but roughly 50 different genes in a typical flowering plant genome [17,33]. This increase in gene number parallels the evolution of body plan complexity in the sporophytes of land plants, in line with the view that the diversification of MIKC-type genes contributed to this in a causal way [34].

In flowering plants, MIKC*-type genes are mainly involved in male gametophyte development, whereas MIKC^C^-type genes are involved in sporophyte ontogeny [17]. The most iconic function of MIKC^C^-type MTFs (M^C^-MTFs) is in the specification of the identity of floral organs, such as petals, stamens and carpels [35,36]. The family of MIKC^C^-type genes includes 12 and 17 well-defined clades that had already been established in the stem group of extant seed plants and flowering plants, respectively [37]. Often, members of the same clade share very similar and conserved functions in diverse developmental processes, such as the *DEF*- and *GLOBOSA*- (*GLO*-) like genes that specify petal and stamen identity, and the *AG*-like genes that specify stamen and carpel identity [3,17,23,34,35,36].

## 3. MIKC Blessing 2.0: A Prayer in C

M^C^-MTFs are typical transcription factors on many accounts. However, there is one feature that sets them apart from almost all other transcription factors and transcriptional regulators—their eager formation of heterotetrameric complexes that bind as dimers of dimers (i.e., a tetramer) to two CArG boxes on the same strand of DNA, requiring the bending of the DNA between the binding sites [11,38,39]. Tetramerization of MIKC-type proteins was first identified by analyzing the mode of action of proteins that specify the identity of floral organs in angiosperms, resulting in the floral quartet model (FQM) [11,38,39,40]. Later, it was found that the formation of such protein complexes is a more general feature of M^C^-MTFs, reaching beyond the formation of floral quartets; accordingly, the term floral quartet-like complexes (FQCs) was coined for all such DNA-bound tetramers, whether they are involved in flower development or not [11].

FQC formation depends on some remarkable structural features of the K domain [7], but not all MIKC-type proteins can accomplish this. Recent data suggest that some MIKC-type proteins of charophyte algae are capable of FQC formation, but that an exon duplication that led to an elongation of the K domain in the stem group of extant MIKC^C^-type genes strongly favored it [31]. In contrast, MIKC*-type proteins appear to bind to DNA only as dimers, not as tetramers [31].

Tetramerization of proteins involved in transcriptional regulation, and binding to two sequence elements involving DNA looping, is well known from bacterial repressors and activators, such as the lac repressor and lambda repressor/activator [41,42,43]. It has also long been known that MADS-domain proteins act in multimeric complexes. However, in cases other than M^C^-MTFs, dimers of MADS-domain proteins form complexes with proteins that are not members of the MADS-domain protein family, such as homeodomain or HMG-domain proteins [2,44]. Tetrameric complexes composed exclusively of MADS-domain proteins (encoded by the same or paralogous genes) appear to be unique to M^C^-MTFs. We hence consider the tetramerization of M^C^-MTFs and FQC formation as important evolutionary novelties in gene regulation. We believe that these insights could help to solve an important conundrum regarding the target-gene specificity of M^C^-MTFs, which has been debated for decades: how can dozens of very similar and highly related (paralogous) transcription factors that recognize very similar DNA sequences (including but not limited to ‘perfect’ CArG boxes) that occur hundreds of thousands of times in a typical flowering plant genome accurately recognize their target genes?

The available evidence indicates that there is an interplay of different mechanisms at work, collectively constituting a ‘floral quartet code’ of target site recognition (Figure 2). In the following, we focus on the major mechanisms involved that have been recognized so far. In Section 4.1 and Section 4.2, we discuss the different types of DNA binding, i.e., DNA contacts in the major vs. the minor groove involving base and shape readout. In Section 4.3 and Section 4.4, we focus on special requirements for the CArG-box sequence and the potential length of the binding motif. In Section 5, Section 6 and Section 7, we review the role of the MADS, I and K domains for protein–DNA and protein–protein interactions, the role of the dimerization and tetramerization of M^C^-MTFs, cooperative DNA binding to two CArG boxes and the optimal CArG-box distance and orientation.

## 4. Recognition of DNA-Sequence Elements by MADS-Domain Proteins

### 4.1. Base Readout

As for most transcription factors, also MTFs utilize the mechanism of base readout to identify target sequences. Base readout, also termed direct readout, describes the recognition of the DNA sequence by protein–DNA contacts mainly in the major groove of the DNA [45]. This works through interactions of the amino acid side chains of the transcription factor via hydrogen bonds or hydrophobic interactions with the bases or base pairs of the DNA [45]. The result is the preference for specific nucleotides at specific positions of the motif.

The crystal structures of the DNA-binding MADS domain and the intervening domain of the M^C^-MTF SEPALLATA3 (SEP3) have recently been elucidated, but without bound target DNA [6]. Therefore, we still rely on modelling [46] and on available X-ray crystallography and NMR structures of human and yeast MTFs MCM1, MEF2A, MEF2B and SRF [8,9,10,47,48] to assess the protein–DNA contacts of plant MTFs.

According to available crystal and NMR structures of protein dimers of the human MTFs SRF and MEF2A bound to their target DNA, protein–DNA contacts are made with both the minor and the major grooves of the target DNA [8,9,10,47]. The N-terminal arm, the α-helix H1 and the β-hairpin loop (the latter is only true for SRF) of the MADS domain of each monomer are involved in DNA binding.

The α-helix H1 interacts with the major groove and the phosphate backbone and makes base-specific contacts predominantly at the edge of the 10-base-pair CArG-box sequence and beyond [8,9,10,47]. Protein–DNA contacts between one lysine residue of the α-helix of each monomer and the two guanine residues on each DNA strand (5′-CC(A/T)_6_**GG**-3′ and 3′-**GG**(A/T)_6_CC-5′; guanine residues on both strands are marked in bold) in the major groove of the target DNA are responsible for the requirement of the ‘CC’ and ‘GG’ borders of the CArG-box motif [8,47]. Amino acid residues of the α-helix and the β-loop make hydrophobic contacts with one (SRF) or two thymine residues (MCM1), respectively, in the flanking regions of the CArG box [8,47].

The A/T-rich CArG-box center is bound mostly in the minor groove by the MTF, although DNA contacts in the major groove also exist [8,9,10,47]. Overall, in the case of MTFs, base readout is especially important to identify the ‘CC’ and ‘GG’ borders of the CArG-box motif and, to a lesser extent, also to recognize the A/T-rich CArG-box center and the flanking sequences.

### 4.2. Shape Readout

The presence of a single CArG-box consensus sequence motif 5′-CC(A/T)_6_GG-3′ as a *cis*-element in a regulatory region of a gene is by itself a poor predictor of target gene specificity as it can be found several thousand times in plant genomes, e.g., over 17,000 copies were identified in the genome of the model plant *Arabidopsis thaliana* [14]. Considering that also CArG boxes with one mismatch compared to the consensus sequence can be functionally relevant in vivo, almost all genes in the *A. thaliana* genome have a potential binding site for MTFs [14]. Additionally, the MIKC-type MTF family is a large family in plants, encoded by approximately 40 genes in *A. thaliana* [23,28,49]. Almost all of the MTFs need to recognize specific target genes; otherwise, developmental processes may run havoc, as exemplified by homeotic mutants in which organ identities are changed [5,35,36]. How is this achieved against all these odds?

One means by which MTFs have increased sequence specificity is the shape readout of the target DNA [45,50,51,52,53]. Shape readout, also termed indirect readout, refers to the recognition of the sequence-dependent three-dimensional structure and the deformability of the DNA by DNA-binding proteins [45,54]. One well-described type of shape readout is the recognition of the minor groove width [54]. Depending on the DNA sequence, several DNA shape parameters, including the minor groove width, can vary greatly. Very narrow minor grooves of the DNA occur especially when so-called A-tract sequences are present. A-tracts are A/T-rich sequences with the special feature of having at least four consecutive A·T base pairs without an intervening TpA step, i.e., A_n_T_m_ with n + m ≥ 4 [55,56].

According to available 3D structures of protein–DNA complexes of SRF and MEF2A, protein–DNA contacts within the A/T-rich CArG-box center are made primarily by amino acid residues of the N-terminal extension in the DNA minor groove [8,9,10]. In addition, some contacts are provided by α-helix H1 with the DNA backbone. Therefore, the N-terminal extension seems to be the major determinant of minor groove shape readout.

Several studies investigated the DNA-binding mechanism of MTFs employing SELEX-seq (Systematic Evolution of Ligands by EXponential Enrichment DNA-Sequencing), an in vitro selection method, which starts with a random DNA library and yields high-affinity DNA-binding sequences for the studied protein, and ChIP-seq (Chromatin ImmunoPrecipitation DNA Sequencing). The studies revealed that the narrow minor groove recognition of A-tract sequences within the A/T-rich CArG-box core (5′-CC(A/T)_6_GG-3′ in the case of SRF-type CArG-boxes) is an important DNA-binding mechanism of MTFs [45,51,52,53,57]. It has been shown that at least some (and maybe most) MTFs, e.g., AG, APETALA 1 (AP1), APETALA3 (AP3), FLOWERING LOCUS C (FLC), MEF2B, PISTILLATA (PI), SEP3, SUPPRESSOR OF OVEREXPRESSION OF CONSTANS1 (SOC1) and SHORT VEGETATIVE PHASE (SVP), preferentially bind CArG boxes containing A-tract sequences over non-A-tract sequences [50,51,52,53,58].

The preference for A-tract-containing CArG boxes obviously limits the number of potential binding sites since only 36 out of the 64 CArG-box sequences with the consensus sequence 5′-CC(A/T)_6_GG-3′ contain an A-tract (e.g., 5′-CCAAATTTGG-3′, but not 5′-CCTTTAAAGG-3′). Additionally, stretches of three to six consecutive adenines within the CArG box (or thymines on the reverse strand), i.e., AAA, AAAA, AAAAA or AAAAAA, are often preferred [58]. In particular, 31 out of 64 SRF-type CArG boxes fulfill both criteria: the presence of an A-tract and at least three consecutive adenines. Hereby, different M^C^-MTFs seem to prefer A/T-rich sequences or A-tracts, respectively, of different lengths [52,58].

The importance of shape readout for MTFs was also shown by demonstrating that the prediction of DNA-binding events based only on the CArG-box DNA sequence was not satisfactory [50,51,53,59], because these predictions depend on the assumption of independent protein–DNA interactions for each DNA base pair of the binding motif. Instead, modelling approaches of DNA-binding events using mixed models of DNA sequence and DNA shape parameters were superior to models based on the DNA sequence alone [51,59]. DNA shape parameters are important because they include information on neighboring base pairs for each base pair and thus on special DNA conformations, which are, e.g., present in A-tract sequences. Alternatively, modelling approaches, which include information on the dependency of different positions within the CArG-box motif, work better than simple models assuming the independence of positions [57,60].

### 4.3. Differences in DNA-Binding Specificity

The combination of base and shape readout enables each MTF to specifically bind only to a (more or less unique) subset of sequences of the canonical CArG-box motif. A review of several ChIP-seq studies revealed subtle differences in the consensus sequence for different MIKC-type MTFs [58]. However, the consensus sequence for each transcription factor is not sufficient to describe the DNA-binding behavior of MTFs. Instead, looking more carefully at ChIP-seq scores [53] or ChIP-seq score means [50] reveals that different MTFs bind each ‘perfect’ CArG box with a different affinity. This means that a certain CArG box with the consensus sequence 5′-CC(A/T)_6_GG-3′ can theoretically be bound by different MTFs; however, most likely, in vivo, it will only be bound by the MTFs for which it has a high (and not a low) DNA-binding affinity.

Gel electrophoretic mobility shift assays (EMSAs) [50] and SELEX-seq studies [52,57] confirmed that there are quantitative differences in DNA-binding affinities for different CArG boxes by SEP3 homodimers. Smaczniak et al. studied different dimer complexes of MIKC-type MTFs (homo- and heterodimers) in a SELEX-seq study and showed that different protein dimers bound DNA probe sequences with different specificities and affinities [52].

Lai et al. have recently presented the newly developed method seq-DAP-seq (sequential DNA-affinity purification sequencing) [61]. This method can give insights into complex-specific binding since it can separate homomeric and heteromeric protein complexes. The authors used it successfully to show that the DNA-binding specificity of SEP3 homomeric and SEP3-AG heteromeric complexes on genomic DNA differs.

On the one hand, ChIP-seq and other studies indicated that MIKC-type MTFs have largely overlapping target genes, can act as transcriptional repressors as well as activators and can interact with different cofactors to regulate target gene activity, while, on the other hand, ChIP-seq studies revealed that there are also distinct target genes for the different MIKC-type proteins [62,63,64,65,66,67]. It is therefore still challenging to assess which role the DNA-binding specificity plays in vivo in the context of achieving target gene specificity.

### 4.4. Length of the DNA-Binding Motif

Considering the length of the DNA-binding motif, there are a number of reports indicating that the binding site for MIKC-type proteins might be considerably longer than the canonical 10-bp-long SRF-type CArG box. SELEX binding site enrichment experiments of AG, SHATTERPROOF1 (SHP1, previously known as AGL1), SEPALLATA1 (SEP1, AGL2) and SEPALLATA4 (SEP4, AGL3), which were conducted almost three decades ago, indicated that three nucleotides on either side of the CArG box are also part of the binding motif [68,69,70,71] (Figure 3A–E). Similar results were also found for SRF [13] (Figure 3F). Therefore, the DNA-binding motif might be rather 16 bp long instead of only 10 bp. Remarkably, these 3-bp flanking sequences were found to be A/T-rich, similar to the CArG-box central motif.

ChIP-seq studies on AG [64], AP1 [63,65], AP3 [67], FLC [74,75], FLM [76], PI [67], SEP3 [65,77], SOC1 [78] and SVP [74,79] largely corroborated this view. Aerts et al. re-analyzed some of these data sets and concluded that a short 3-bp-long A/T-rich extension on one side of the CArG box, namely 5′-NAA-3′ on the 3′-side, is important for DNA binding [58].

More recently, two SELEX-seq studies provided deeper insights into the DNA-binding specificities of AP1, AG and SEP3. Both studies found that 5′-TTN-3′ (at the 5′-end) and 5′-NAA-3′ (at the 3′-end) were the prevalent flanking sequences of the CArG-box motif [52,57] (Figure 3G–O).

To summarize, SELEX, ChIP-seq and SELEX-seq studies found that the DNA-binding motif of M^C^-MTFs is 16 bp rather than 10 bp long. However, there were high sequence similarities in the flanking sequences between different transcription factors. Although longer binding motifs limit the number of potential binding sites and hence the number of functional CArG-box sequences in the genome, the flanking sequences might rather help to differentiate between functional and non-functional CArG boxes instead of conferring DNA-binding specificity within the M^C^-MTF family.

## 5. The Role of the Protein Structure

### 5.1. The General Contribution of MADS and I Domain to Target Gene Specificity

The molecular functions of the MADS and the I domain for DNA binding and dimerization are largely understood. A truncated MADS-domain protein containing only the MADS and the I domain is usually able to dimerize and to bind DNA in a sequence-specific manner—a few exceptions not withstanding [6,50,69]. How the DNA-binding specificity on the one hand and target and functional specificity on the other hand are related has been under debate for decades [80,81,82].

Reports from domain-swap experiments between different MTFs [81], as well as structural information of yeast and human MTFs [8,9,10,47,48], indicated that the in vitro DNA-binding specificity resides in the N-terminal half of the MADS domain.

However, if hybrid proteins with the complete MADS domain from one protein and the I, K and C domains from another floral homeotic protein were ectopically expressed in planta, the overexpression phenotype was largely determined by the identity of the non-DNA-binding part of the hybrid protein, i.e., mainly by the I and the K domain [82]. In a similar study, chimeric transcription factors were created by substituting the N-terminal half of the MADS domain of plant MTFs with the corresponding sequence of the human MTFs SRF or MEF2A, respectively [81]. The overexpression phenotype of these constructs in planta was dependent on the plant transcription factor identity and independent of the protein identity providing the N-terminal half of the MADS domain. These and other studies were taken as evidence that DNA-binding specificity plays only a minor role in achieving target gene specificity and that additional cofactors are involved in recruiting different floral homeotic proteins to different targets, or that the regulatory activity (i.e., activation or repression) at a particular target gene differs for different floral homeotic proteins [64,80,81].

A recent study re-examined the importance of the I domain for DNA binding, dimerization and in planta transcription factor function [6]. Lai et al. showed that the I domain is required for DNA binding, although there are no direct contacts between the I domain and DNA. Constructs made up only of the MADS domain could interact in pull-down assays, but could not bind to DNA in the absence of the I domain in EMSA experiments. The protein–protein interaction of MADS domains seems to be relatively weak and the I domains seem to be essential to stabilize dimerization, which in turn is a prerequisite for the DNA binding of MTFs. Lai et al. also showed that an I-domain swap between different M^C^-MTFs affected DNA-binding specificity. They postulated that the reason for this is an allosteric effect of the I domain that influence the MADS-domain conformation and thereby tune the DNA-binding specificity [6].

Lai et al. (2021) also used I-domain swaps to show that the I-domain identity is important for dimerization specificity in vitro and in yeast two-hybrid screens. I-domain-swap experiments in planta indicated that the I domain is the major determinant for the successful complementation of MTF function in a mutant background [6]. These recent in planta studies are in good agreement with the aforementioned studies from the 1990s [81,82].

However, the study conducted by Lai et al. (2021) can also help to reconcile the dispute about the importance of the MADS domain and of DNA-binding specificity for target gene specificity. According to their results, the MADS domain provides the DNA contacts and is therefore important for the general recognition of CArG-box sequences by the MTF family. The specificity in terms of binding only MTF-specific CArG boxes and target genes seems to be provided to a large extent by the I domain as it appears to be able to modulate DNA-binding specificity through allosteric effects on the MADS domain.

### 5.2. What Single Amino Acid Substitutions in the MADS and I Domain Tell Us

Several studies on single amino acid substitutions in the MADS and the I domain have been conducted in the past. Some of these studies have extended the understanding of the protein–DNA interactions and have identified critical amino acid residues for DNA binding (specificity).

The substitution of lysin (K) at amino acid position 4 by glutamic acid (E) (in short, K4E) in the N-terminal extension of the MADS domain of MEF2B strongly diminished DNA binding in EMSA experiments [83]. Additionally, only very few ChIP-seq peaks could be detected for the MEF2B K4E mutant in comparison with the wild-type MEF2B [83]. MEF2B K4E was also examined in a SELEX-seq study [51]. In this study, the authors found that the DNA-binding preference was generally similar to that for the wild-type MEF2B; however, MEF2B K4E showed a lower preference for DNA sequences that deviate from the MEF2B consensus binding motif [51]. Similar results were obtained for the N-terminal mutant MEF2B K5E [51].

Two other studies focused on the N-terminal extension mutants R3A and R3K of SEP3, in which the arginine at position 3 was substituted by alanine or lysine, respectively [50,57]. These studies showed that the highly conserved arginine residue R3 is important for DNA-binding affinity and specificity. R3 confers the shape readout of A-tract sequences within the A/T-rich CArG-box core [50]. The SELEX-seq study on SEP3 R3A and SEP3 R3K showed that the binding motif of the mutants compared to the SEP3 wild type differed mostly at the A/T positions directly 3′ of the ’CC’ and 5′ of the ’GG’ CArG-box borders, which means that the recognized A/T-rich core is only four base pairs long for the mutants, instead of six base pairs for the wild type (5′-CC(A/T)_6_GG-3′) [57].

Crystal structures of human MADS-domain proteins strongly suggested that the N-terminal arm of the MADS domain is involved in DNA minor groove binding [8,9,10]. The experimental results for arginine R3 of SEP3 were in good agreement with the hypothesis that arginine residues are employed for the minor groove shape readout of A-tract sequences by several transcription factor families, among them also MTFs [45,54].

Two α-helix H1 mutants of MEF2B, MEF2B R15G (arginine at position 15 substituted by glycine) and MEF2B K23R (lysine at position 23 substituted by arginine) were also part of the aforementioned SELEX-seq study [51]. MEF2B R15G and MEF2B K23R are known from MEF2 structures to contact the DNA in the flanking sequences or at the CArG border, respectively [10,48]. Both mutants showed a larger shift in DNA shape preference compared to wild-type MEF2B than observed for the N-terminal extension mutants [51]. The most obvious change in the DNA-binding motif for MEF2B K23R was the loss of the 5′ cytosine and the 3′ guanine of the consensus CArG-box motif, which can easily be explained by the loss of direct DNA contacts with the 3′ guanine of each MEF2 monomer with each DNA strand [10].

Lei et al. have solved the crystal structure of a MEF2A/MEF2B chimera with the mutation D83V in the MEF2 domain [84], which is functionally very similar to the I domain. This amino acid substitution leads to a structural change in the MEF2 domain, whereby the α-helix H3 is switched into the beta strand β4. In the wild-type MEF2B, helix H3 contributes to DNA binding in two ways: directly by providing a cluster of positively charged residues towards the DNA surface and indirectly by stabilizing the DNA-contacting α-helix H1 of the MADS domain. The dissolution of helix H3 seems to have modest effects on DNA binding [84]. This is in agreement with another study using EMSA and ChIP-seq experiments and showing that MEF2B D83V has a lower DNA-binding affinity and fewer ChIP-seq peaks, respectively, than the wild-type MEF2B [83].

SEP3 I-domain mutations R69L, R69P and Y70E, which lie within the α-helix H2, destabilized SEP3 according to thermal shift assays [6]. In addition, these mutations abolished the DNA binding of SEP3 in an EMSA experiment [6]. Since R69 and Y70 are seemingly important for the structural stability of the I domain and of the whole MTF, effects on DNA binding can be seen. These studies on MEF2B and on SEP3 [6,83,84] support the notion that the MEF2 domain or the I domain, respectively, can allosterically influence the DNA-binding behavior of MTFs while possessing no direct DNA contact.

### 5.3. The Keratin-Like Domain—Mediator of Tetramerization

MIKC-type MTFs are characterized by the presence of the keratin-like domain (K domain), a protein–protein interaction domain that shows sequence similarity to the eponymous filament protein keratin (Figure 1A) [19,33,85]. The amino acid sequence within the K domain follows a characteristic pattern of hydrophobic and charged residues that repeats every seven amino acids [86,87,88]. In this so-called heptad repeat pattern of the form [abcdefg]_n_, the a and d positions are mainly occupied by hydrophobic amino acids such as leucine, isoleucine or methionine, and the e and g positions are predominantly occupied by the charged amino acids lysine, arginine, aspartate and glutamate [89,90]. Amino acid stretches that follow this type of heptad repeat pattern are well known from other protein–protein interaction domains, particularly coiled coils and leucine zippers [91,92,93]. Due to the regular spacing of hydrophobic residues in a heptad repeat, the amino acid strand winds up to an amphipathic α-helix, where all hydrophobic residues are directed to the same side of the helix. This way, a hydrophobic stripe is formed that runs around the helix and allows for hydrophobic interactions with one or several other amphipathic α-helices. The charged residues on the heptad repeat e and g positions flank the hydrophobic stripe and mediate additional attractive or repulsive electrostatic interactions [89,90].

The determination of the X-ray crystal structure of the K domain of SEP3 revealed that the K domain indeed folds into two amphipathic α-helixes that are separated by a rigid kink, which prevents the intramolecular interaction of both helixes (Figure 1C) [7]. The first (N-terminal) K-domain helix contains an interaction interface that strengthens the protein–protein interaction of a DNA-bound SEP3 dimer. The N-terminal half of the second (C-terminal) K-domain helix contains a second dimerization interface, whereas the C-terminal half of the second helix harbors a tetramerization interface that facilitates the interaction of two DNA-bound SEP3 dimers and thus FQC formation (Figure 1D) [7]. Although the K domain of SEP3 remains the only one for which structural data are available, analyses of amino acid conservation on interacting sites suggest that the overall structure of the K domain is highly conserved, at least among the M^C^-MTFs of seed plants [88].

Within the SEP3 homotetramer, dimerization and tetramerization are mainly mediated by the strong hydrophobic interactions of leucine residues on heptad repeat d positions [7,88,94]. Salt bridges between glutamic acid/aspartic acid and arginine/lysine residues on the heptad repeat a, e and g positions, respectively, further stabilize dimerization as well as tetramerization [7,88]. Thus far, no structural information is available for side chain interactions in heterodimers or -tetramers of different MTFs. However, interactions of two or more amphipathic α-helices have been intensively studied and it is well known that complex ‘knobs-into-holes’ side chain packing determines the interaction strength and specificity [90,95,96]. It thus appears likely that the presence or absence of hydrophobic and charged residues at critical amino acid positions within the K domain determines whether a certain heterodimer or -tetramer can be formed or not [88]. In a number of studies, amino acid positions within the K domains of the floral homeotic B class proteins AP3 and PI from *A. thaliana* have been identified, which contribute to the obligate heterodimerization of AP3/PI [86,87,97]. Interestingly, reassessment of these amino acid positions in the context of the K-domain structure suggests that attractive or repulsive electrostatic interactions at interacting sites indeed facilitate or impede heterodimer formation [98].

## 6. Origin and Evolution of FQCs

Members of different subfamilies of M^C^-MTFs have been shown to considerably differ in their protein–protein interaction capabilities. Based on large-scale yeast-two-hybrid and yeast-three-hybrid screens of MIKC-type MTFs from *A. thaliana*, some proteins, such as the floral homeotic E class protein SEP3, have been identified as interaction hubs, whereas others, such as the B class proteins AP3 and PI, revealed a very limited set of interaction partners [99,100]. Similar interaction studies have been performed for floral homeotic proteins from other core eudicot species [101,102,103,104,105], early diverging eudicots [106], monocots [107,108,109] and early diverging angiosperms [110,111,112], as well as for orthologs of floral homeotic proteins from gymnosperms [113,114]. Comparisons of the determined protein–protein interaction networks have shown that all interactions required for the formation of the different floral quartets are highly conserved [106,111]. However, in addition to the conserved interactions, floral homeotic proteins from early diverging angiosperms, as well as their orthologs from gymnosperms, show more promiscuous interaction patterns [111,113]. This observation, together with reconstructions of ancestral states of the protein–protein interaction network (PPI) of floral homeotic proteins, suggest that the PPI evolved from a promiscuous ancestral state to a network with increased specificity, with mainly those interactions being retained that are required for formation of the different floral quartets [101,106,111,115].

The formation of floral quartets has been demonstrated in vitro for floral homeotic proteins from *A. thaliana* and the early diverging angiosperm *Amborella trichopoda* [88,116,117], as well as for orthologs of floral homeotic proteins from the gymnosperm *Gnetum gnemon* [113]. Furthermore, analysis of the PPI topology of M^C^-MTFs of *A. thaliana* suggests that also M^C^-MTFs other than floral homeotic proteins can be incorporated into floral quartet-like complexes [118]. Thus, it appears likely that FQC formation is widespread, at least among MIKC^C^-type proteins of seed plants, and likely was already present in a common ancestor of angiosperms and gymnosperms more than 300 million years ago (Ma) [113,115]. Recent data on protein–protein and protein–DNA interactions of MTFs from non-seed plants demonstrated that also M^C^-MTFs from ferns, lycophytes and mosses are capable of forming FQCs, whereas seed plant MIKC*-type proteins as well as most MIKC-type proteins from charophyte green algae (land plants’ closest living relatives) bound to DNA only as dimers [31]. Based on in silico and in vitro analyses, it is hypothesized that the duplication of the last K-domain exon of an ancestral MIKC^C^-type gene that occurred in the stem lineage of extant land plants was the crucial step that elongated the second K-domain helix and thereby gave rise to the tetramerization interface found in extant M^C^-MTFs [31].

## 7. Why Quartets and FQCs?

Now, we have reached the final, but arguably the most intriguing, question: given that so many transcription factors, including MADS-domain proteins, happily work as dimers, why do many (if not all) M^C^-MTFs form tetrameric complexes and FQCs?

Since tetramers of M^C^-MTFs, to begin with, bind to two sites on the DNA, the distance and orientation of CArG boxes affect the strength of DNA binding (Figure 2). It was shown that different tetramers have different DNA-binding affinities and that different tetramers prefer different CArG-box distances for maximum binding [116,119]. These distances between the CArG boxes are surprisingly short, only a few helical turns of the DNA [119]. FQC formation works best if the CArG boxes are in the same orientation because the DNA between them has an integer number (usually 3–7) of helical turns (Figure 2). If they are in an opposite orientation, besides bending, also the twisting of DNA is required, which diminishes binding [119]. By preferring optimal pairs of CArG boxes, FQC formation could thus contribute to an increase in target gene specificity. This offers the possibility to differentially regulate target genes even in the absence of the differential DNA binding of M^C^-MTF dimers [11].

An important difference between two dimers binding independently to DNA and one tetramer binding is, under certain conditions, an increase in cooperativity in DNA binding. This cooperativity can create a sharp transcriptional response, which means that only small increases in the concentration of M^C^-MTFs can lead to drastic changes in the effect on target genes and hence regulatory output [11]. M^C^-MTFs often act as genetic switches that control discrete developmental or physiological stages. Cooperative tetramer formation of M^C^-MTFs on DNA might thus be one important mechanism that translates the quantitative nature of biomolecular interactions into discrete phenotypic outputs [120,121].

Tetramer formation may also incorporate different signals and thereby increase the robustness of the gene regulatory decision on M^C^-MTF target gene expression. If one protein component of the tetramer is missing, the entire complex will not form or will be greatly destabilized, and the developmental switch will not occur [11].

## 8. Conclusions and Outlook

Dozens of similar M^C^-MTFs need to accurately choose their sets of target genes out of hundreds of millions of possibilities in plant genomes; otherwise, serious developmental abnormalities may occur. A number of mechanisms involved are meanwhile quite well understood and have been outlined in this review, comprising the base and shape readout of individual CArG boxes by M^C^-MTF dimers, dimerization specificity determined by amino acid sequence features within the I and K domains, the presence of suitably oriented pairs of CArG boxes and the ability to cooperatively bind to two CArG boxes by forming M^C^-MTF tetramers (Figure 2). The potential role of non-M^C^-MTF interaction partners has recently been reviewed [12] and has hence not been considered here. Some mechanisms of target gene binding involving the chromatin structure (Figure 2) have been proposed or reviewed previously [11] but are still highly speculative and are hence also not covered here. There is no guarantee, however, that even all these different mechanisms together will eventually be sufficient to decipher the floral quartet code and explain the impressive functional specificity of M^C^-MTFs. Some other mechanisms not on the agenda of MADS research so far might be required for a comprehensive explanation. It is possible that they are forthcoming—we encourage readers to remain alert.

## Figures and Tables

**Figure 1 ijms-24-08253-f001:**
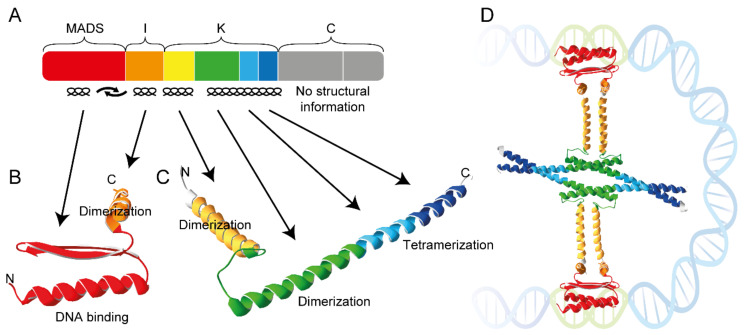
Domain architecture and structure of the MTF SEP3 from *A. thaliana*. (**A**) Colored boxes represent the eight exons that encode SEP3 from *A. thaliana*. Exon one (red) encodes the DNA-binding MADS domain, exon two (orange) the intervening domain (I), exons three to six (yellow, green, light blue and dark blue) the keratin-like domain (K) and exons seven and eight (grey) the C-terminal domain (C). Secondary structures of the encoded domains are indicated below the colored boxes. (**B**) X-ray crystal structure of the MADS domain (red) and I domain (orange) of SEP3 (PDB-ID: 7NB0) as determined by [6]. (**C**) X-ray crystal structure of the K domain of SEP3 (PDB-ID: 4OX0) as determined by [7]. Subdomains encoded by exons three to six follow the color coding of panel A. (**D**) Hypothetical composite structure of a SEP3 homotetramer forming an FQC. The N-extension of the MADS domain that contacts the DNA minor grove is not covered by the structure shown in panel (**B**).

**Figure 2 ijms-24-08253-f002:**
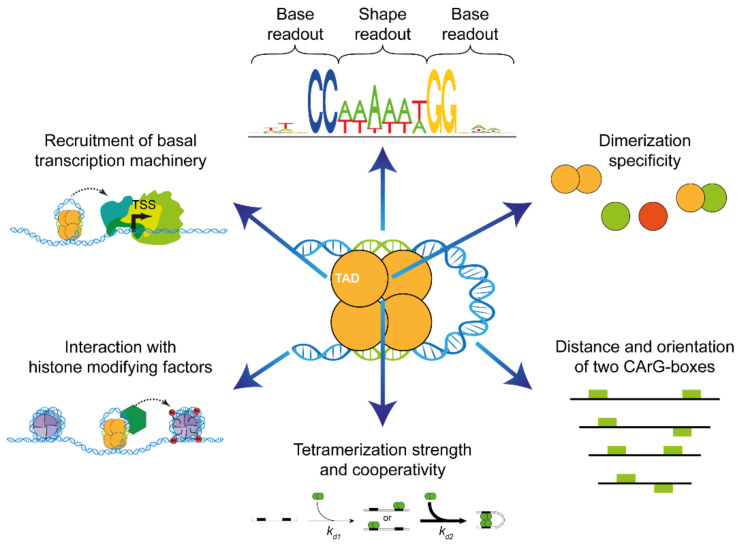
Towards an FQC code of target gene recognition (from top in a clockwise direction). A single CArG box and its flanking regions are recognized by a MTF dimer via a combination of base and shape readout. Attractive or repulsive forces between the dimerization interfaces of two interacting MTFs facilitate or impede dimerization. The distance between two neighboring CArG boxes and whether both are directed to the same site of the DNA double helix determine whether FQC formation is favored or not. Ability to form tetramers facilitates cooperative binding of a second MTF dimer while looping the DNA in between both binding sites. Pioneering MTF tetramers may compete with histones or recruit histone-modifying factors [11]. Presence of at least one transactivation domain (TAD) in a DNA-bound MTF tetramer recruits the basal transcription machinery and eventually initiates transcription at the transcriptional start site (TSS). The important aspect of co-factor binding to MTFs is largely neglected here, because it has been covered comprehensively in a recent review already [12].

**Figure 3 ijms-24-08253-f003:**
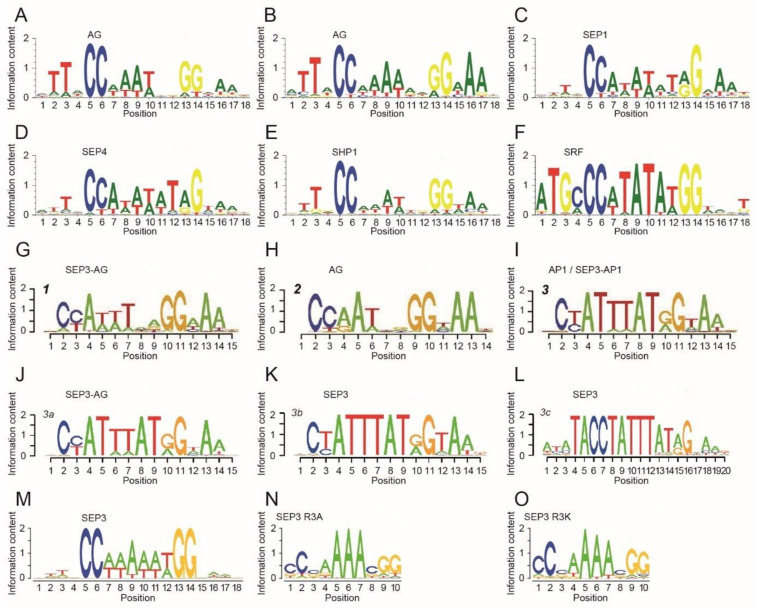
Binding motifs of different MTFs determined by SELEX and SELEX-seq. (**A**–**F**) Binding motifs of the MIKC-type MTFs AG, SHP1, SEP1 and SEP4 from *A. thaliana* and SRF from human as determined by low-resolution SELEX experiments [13,68,69,70,71]. Sequence logos were generated with Weblogo3 [72,73] based on the position weight matrices determined in the individual studies. (**G**–**L**) Binding motifs of homo- and heterodimers of the MIKC-type MTFs AP1, AG and SEP3, as determined by high-resolution SELEX-seq experiments [52]. (**M**–**O**) Binding motifs of (**M**) SEP3 wild-type protein and the single amino acid substitution mutants (**N**) SEP3-R3A and (**O**) SEP3-R3K as determined by high-resolution SELEX-seq experiments [57].

## Data Availability

Not applicable.

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
