# Peer review of "Cracking the Floral Quartet Code: How Do Multimers of MIKCC-Type MADS-Domain Transcription Factors Recognize Their Target Genes?"

_ijms, 2023, doi:10.3390/ijms24098253_

Round 1

Reviewer 1 Report

the idea is great and it sound good

but i still have some comments

plz rewrite the abstract

in the introduction part plz show us how is the important of your work to biological process

plz make your figs is more clear

check your references

Author Response

Reviewer 1: the idea is great and it sound good, but i still have some comments, plz rewrite the abstract, in the introduction part plz show us how is the important of your work to biological process, plz make your figs is more clear, check your references.

Our response: We are grateful that the reviewer likes our idea and sound, and we highly appreciate all his/her efforts. However, since we did not get any specific clue what kind of problems the reviewer has identified and exactly how we should improve our manuscript, we are desperate to say that we were not able to change the manuscript based on these comments. We feel that the importance of our work on DNA-binding by MTFs to the biological processes of plant development and evolution pervades our manuscript from the Abstract to the Conclusions and outlook. Pointing even more to it some readers might consider as penetrant.

Reviewer 2 Report

Dear authors,

It was with pleasure that I read your manuscript entitled ‘Cracking the floral quartet code: How do multimers of MIKCC-type MADS-domain transcription factors recognize their target genes?’

The structure of the manuscript is rather well thought off, starting with some background and evolutionary history in the MADS family of transcription factors, before zooming in on the MIKC type. The rest of the manuscript is focussing on different topics, all revolving around how MIKC MTFs acquire target specificity. It might be too much to ask, since obviously a lot is still unknown, but the question asked in the title would warrant a paragraph in which it is attempted to answer it, for example by briefly summarizing the findings in the review? Below I outlined some minor remarks I think should be addressed, but I believe the manuscript is fit for publication after these minor revisions have been addressed. 

Line 37-38 ‘Their mode of DNA-binding is a crucial aspect of the mechanism by which they recognize target genes and is hence of great biological interest’ Probably a matter of semantics, but isn’t this true for all transcription factors, not just MADSs?

Line 100-102 ‘In any case, with its very high birth and death rates the evolutionary dynamics of Type I genes of plants deviates considerably from that of both, animal and fungal Type I and plant Type II genes’ – Please elaborate, to what/who do the authors refer to about the high birth and death rates, and how does this relate to the evolutionary dynamics?

The following lines 105-106 and onwards ‘For quite a while all known Type II MTFs of plants had a characteristic and unique structure comprising four domains…’ I am guessing that the authors don’t imply that the Type II MTFs had the four domain structure and lost it, but rather that our understanding of the classification/terminology has changed because of new insights?

Line 377 and onwards, I am confused by the sentence ‘the I-domain seems to be essential for stable dimerization, which is a prerequisite for DNA-binding of MTFs.’ Where in the line before you mention that constructs only made with the MADS domain can interact, but require the I-domain to bind DNA, this seems contradictory?

In chapter/paragraph 5.3 there are a few commas that can be omitted:

Line 468, after ‘revealed’

Line 476, after ‘one’

Line 487, after ‘likely’

Line 488, after ‘determine’

Line 493, after ‘suggest’

Yours,

Author Response

Reviewer 2: It was with pleasure that I read your manuscript entitled ‘Cracking the floral quartet code: How do multimers of MIKCC-type MADS-domain transcription factors recognize their target genes?’. The structure of the manuscript is rather well thought off, starting with some background and evolutionary history in the MADS family of transcription factors, before zooming in on the MIKC type. The rest of the manuscript is focussing on different topics, all revolving around how MIKC MTFs acquire target specificity.

Our response: We thank reviewer 2 for this kind view.

Reviewer 2, comment 1: It might be too much to ask, since obviously a lot is still unknown, but the question asked in the title would warrant a paragraph in which it is attempted to answer it, for example by briefly summarizing the findings in the review?

Our response: We aimed to merge the different aspects that determine FQC formation graphically in Figure 2 and the corresponding figure legend. We added a brief summary to the Conclusions and outlook section and refer to Figure 2 (lines 571-575): “A number of mechanisms involved are meanwhile quite well understood and have been outlined in this review, comprising base and shape readout of individual CArG-boxes by MC-MTF dimers, dimerization specificity determined by amino acid sequence features within the I- and K-domain, presence of suitably oriented pairs of CArG-boxes, and the ability to cooperatively bind to two CArG-boxes by forming MC-MTF tetramers (Figure 2).”

Reviewer 2, comment 2: Line 37-38 ‘Their mode of DNA-binding is a crucial aspect of the mechanism by which they recognize target genes and is hence of great biological interest’ Probably a matter of semantics, but isn’t this true for all transcription factors, not just MADSs?

Our response: Sure, absolutely! But if you say that “the eyes are a crucial aspect by which a lion recognized its prey”, that doesn’t imply that the tiger doesn’t use its eyes in the same way. However, to avoid misunderstandings, we rephrased the sentence (lines 37-38): “Like for all transcription factors, their mode of DNA-binding is a crucial aspect of the mechanism by which they recognize target genes, and is hence of great biological interest”

Reviewer 2, comment 3: Line 100-102 ‘In any case, with its very high birth and death rates the evolutionary dynamics of Type I genes of plants deviates considerably from that of both, animal and fungal Type I and plant Type II genes’ – Please elaborate, to what/who do the authors refer to about the high birth and death rates, and how does this relate to the evolutionary dynamics?

Our response: We rephrased the sentence to clarify that the high birth and death rates refer to plant Type I genes. We now write (lines 101-104): “In any case, flowering plant Type I genes have experienced faster birth-and-death evolution than Type II MADS-box genes in angiosperms [121] and other plants [18]. They deviate in their evolutionary dynamics also from both animal and fungal Type I genes, in that they originated and got lost more rapidly than the other, highly conserved gene types [18].”

  1. Nam ,J., et al., Type I MADS-box genes have experienced faster birth-and-death evolution than type II MADS-box genes in angiosperms. Proc Natl Acad Sci U S A, 2004. 101(7): p. 1910-5.

Reviewer 2, comment 4: The following lines 105-106 and onwards ‘For quite a while all known Type II MTFs of plants had a characteristic and unique structure comprising four domains…’ I am guessing that the authors don’t imply that the Type II MTFs had the four domain structure and lost it, but rather that our understanding of the classification/terminology has changed because of new insights?

Our response: We thank the reviewer for making us aware of this potential misunderstanding of our statement. We rephrased the sentence to clarify this (lines 105-106): “For quite a while, all Type II MTFs of plants, that had been identified, possessed a characteristic and unique structure comprising four domains, the DNA-binding MADS-domain (M), the intervening I domain (I), the keratin-like domain (K) and the variable C-terminal domain (C) [17, 19, 20].”

Reviewer 2, comment 5: Line 377 and onwards, I am confused by the sentence ‘the I-domain seems to be essential for stable dimerization, which is a prerequisite for DNA-binding of MTFs.’ Where in the line before you mention that constructs only made with the MADS domain can interact, but require the I-domain to bind DNA, this seems contradictory?

Our response: We agree that this phrasing appears contradictory. We rephrased this section to clarify which results were obtained by which kind of assays/methods. We now write (lines 378-382): “Constructs made up only of the MADS-domain could interact in pull-down assays, but could not bind to DNA in the absence of the I-domain in EMSA experiments. The protein-protein interaction of MADS-domains seems to be relatively weak and the I-domains seem to be essential to stabilize dimerization, which in turn is a prerequisite for DNA-binding of MTFs.”

Reviewer 2, comment 6: In chapter/paragraph 5.3 there are a few commas that can be omitted:Line 468, after ‘revealed’, Line 476, after ‘one’, Line 487, after ‘likely’, Line 488, after ‘determine’, Line 493, after ‘suggest’.

Our response: Corrected.

Reviewer 3 Report

Käppel et al. wrote a comprehensive review about the different aspects that together determine the affinity of MADS-domain complexes for distinct target sites. It nicely summarizes the most recent insights on how these different aspects influence specific binding.

The review is well-structured, going from background information (sections 1-3), to the different aspects that influence DNA-binding specificity (sequence, shape (4) and protein structure (5)), to the origin (6) and ‘why’ of floral quartet binding (7).

I think that swapping sections 6 and 7 would improve the text, as the ‘why’ of floral quartet binding (the higher efficiency can be crucial for phase transitions) would follow very well after the two other sections on specificity. The section on when it has evolved can then follow as a separate section or be simply added after the ‘why’ part.

The introductory part (sections 1-3) is quite long. The readability would be improved if it would be shortened a bit (so that the interest of the reader has not already disappeared before the interesting part starts). This part also contains a few sentences that do not read well:

-       Lines 36-37: which other eukaryotes?

-       Line 59: ‘makes’ should be ‘make’

-       Lines 73-74: rephrase

-       Line 105: ‘For quite a while….’  I don’t think that you can say that they had it for quite a while. ‘It was thought for quite a while…..’ . However, to improve the readability, I would delete this part and focus simply on M- and MIKC-type.

-       The title of section 3 is a bit too cryptic for me.

-       Lines 158-159: ‘dimers of dimers’

-       Lines 176-178: rephrase (unclear to me)

-       Sentences can be very long sometimes, and thereby loose significance. For example, the sentence from line 182 to line 186 is important for the rest of the text, but so long that the meaning gets lost a bit. The same for the last sentence of the paragraph (lines 189-195): important, but way too long.

Other minor comments:

-       Line 28: I would not use the word ‘treatment’ here.

-       I would change lines 90-94 about the Type I and Type II lineages (the authors also explain thereafter - that it is not clear whether the Type I in plants are the same as the Type I in animals, but this is suggested by lines 90-94, and the whole explanation makes it difficult to read for non-experts). On bioRxiv, the group of Claudia Köhler now deposited a new paper ‘Updated phylogeny and protein structure predictions revise the hypothesis on the origin of MADS-box transcription factors in land plants’ that sheds more light on it. I would include this information and rewrite the lines about Type I and II. (or don’t make this distinction and focus on the plant MTFs with M-type and MIKC-type).

-       Maybe shortly explain somewhere the principle of SELEX? Finding context of the CArG-box and maybe also shape read-out are likely more difficult with SELEX?

-       About the length of the CArG-box (e.g. lines 341-342): In many PWMs, the extension seems to occur on only one side, so would a length of 13 bp not be better than 16?

-       Line 386: transcription factor function should be MTF function.

-       Lines 393 – 395: I would rather say that the I-domain determines PPI-interaction specificity, and thereby the nature of the dimer, and that the nature of the dimer determines DNA-binding specificity through allosteric effects.

-       Figure 2 is a nice overview that shows all different aspects that contribute to the target site specificity of MTF complexes. The last two aspects (interaction with chromatin modifiers and transcriptional machinery) do not really fit and are not discussed in the text. I would therefore leave these out. Possibly, the authors could add differential binding to additional co-factors (likely via the C-terminus). This most likely also contributes to differential regulation. (instead of showing only target site recognition, it would then be extended to target gene regulation)

Author Response

Reviewer 3: Käppel et al. wrote a comprehensive review about the different aspects that together determine the affinity of MADS-domain complexes for distinct target sites. It nicely summarizes the most recent insights on how these different aspects influence specific binding. The review is well-structured, going from background information (sections 1-3), to the different aspects that influence DNA-binding specificity (sequence, shape (4) and protein structure (5)), to the origin (6) and ‘why’ of floral quartet binding (7).

Our response: We thank reviewer 3 for this positive feedback.

Reviewer 3, comment 1: I think that swapping sections 6 and 7 would improve the text, as the ‘why’ of floral quartet binding (the higher efficiency can be crucial for phase transitions) would follow very well after the two other sections on specificity. The section on when it has evolved can then follow as a separate section or be simply added after the ‘why’ part.

Our response: In the last subsection of 5 (5.3. The keratin-like domain – mediator of tetramerization) we describe the structure of the K-domain as basis for MC-MTF tetramerization. The question on how FQC formation originated and changed during evolution is closely linked to structural and sequence features of the K-domain. Therefore, we would like to stick to the current order of sections 5, 6, and 7.

Reviewer 3, comment 2: The introductory part (sections 1-3) is quite long. The readability would be improved if it would be shortened a bit (so that the interest of the reader has not already disappeared before the interesting part starts).

Our response: It is pretty obvious that the reviewer is an expert in the field. We fully understand that for such readers an Introduction like ours might appear lengthy. We assume, however, that experts like him/her are not representative for the average reader of our paper. Many of these “ordinary people” might appreciate a more detailed Introduction. We would, therefore, like to keep the length as it is.

Reviewer 3, comment 3: Lines 36-37: which other eukaryotes?

Our response: We have extended our statement to answer the question of the reviewer: “These MADS-domain transcription factors (MTFs) play important roles in the development and physiology of plants, animals and fungi, and possibly in almost all other eukaryotes, comprising diverse groups such as ciliates, trypanosomes, radiolarians, and many more, as well”

Reviewer 3, comment 4: Line 59: ‘makes’ should be ‘make’

Our response: We prefer to keep “makes”, because it refers to “layer”.

Reviewer 3, comment 5: Lines 73-74: rephrase

Our response: We rephrased the sentence. We now write (lines 73-74): “How did extant MADS-box genes and MTFs, with their unique fold of the DNA-binding MADS-domain and their recognized cis-regulatory element (CArG-box), originate?”

Reviewer 3, comment 6: Line 105: ‘For quite a while….’  I don’t think that you can say that they had it for quite a while. ‘It was thought for quite a while…..’ . However, to improve the readability, I would delete this part and focus simply on M- and MIKC-type.

Our response: Please see our response to comment 4 of reviewer 2.

Reviewer 3, comment 7: The title of section 3 is a bit too cryptic for me.

Our response: Many readers appreciate it if they get some riddles to solve (except they concern the scientific core of a paper). Just for Reviewer 3: “MIKC blessing” is a pun that refers to the MIKC-domain structure, “a prayer in C” extends the “religious” phrasing by focusing on the C domain, and cites a famous song by Lily Wood & The Prick (just google ;-).

Reviewer 3, comment 8: Lines 158-159: ‘dimers of dimers’

Our response: Changed to “dimers of dimers (i.e. a tetramer)”.

Reviewer 3, comment 9: Lines 176-178: rephrase (unclear to me)

Our response: We thank the reviewer for the kind comment and have rephrased the sentence: “However, in cases other than MC-MTFs, dimers of MADS-domain proteins constitute complexes with proteins that are not members of the MADS-domain protein family, such as homeodomain or HMG-domain proteins [2, 42].”

Reviewer 3, comment 10: Sentences can be very long sometimes, and thereby loose significance. For example, the sentence from line 182 to line 186 is important for the rest of the text, but so long that the meaning gets lost a bit. The same for the last sentence of the paragraph (lines 189-195): important, but way too long.

Our response: We rephrased and shorted these sentences. We now write (line 190-197): “On the following we focus on the major mechanisms involved that have been recognized so far. In chapters 4.1 and 4.2 we discuss the different types of DNA-binding i.e. DNA contacts in the major vs. the minor groove involving base and shape readout. In chapters 4.3 and 4.4 we focus on special requirements for the CArG-box sequence and the potential length of the binding motif. In chapters 5-7 we review the role of the MADS-, I- and K-domain for protein-DNA and protein-protein interactions, the role of dimerization and tetramerization of MC-MTFs, cooperative DNA-binding to two CArG-boxes and the opti-mal CArG-box distance and orientation.”

Reviewer 3, comment 11: Line 28: I would not use the word ‘treatment’ here.

Our response: Changed to “review”.

Reviewer 3, comment 12: I would change lines 90-94 about the Type I and Type II lineages (the authors also explain thereafter - that it is not clear whether the Type I in plants are the same as the Type I in animals, but this is suggested by lines 90-94, and the whole explanation makes it difficult to read for non-experts). On bioRxiv, the group of Claudia Köhler now deposited a new paper ‘Updated phylogeny and protein structure predictions revise the hypothesis on the origin of MADS-box transcription factors in land plants’ that sheds more light on it. I would include this information and rewrite the lines about Type I and II. (or don’t make this distinction and focus on the plant MTFs with M-type and MIKC-type).

Our response: We highly appreciate this view. However, as far as we know, the mentioned paper has not been officially published after peer review yet. We don’t know, therefore, how reliable the information in that paper is and would thus prefer not to include it into our review.

Reviewer 3, comment 13: Maybe shortly explain somewhere the principle of SELEX? Finding context of the CArG-box and maybe also shape read-out are likely more difficult with SELEX?

Our response: We added a short explanation of SELEX when we first mention the method (lines 257-261): “Several studies investigated the DNA-binding mechanism of MTFs employing SELEX-seq (Systematic Evolution of Ligands by EXponential Enrichment DNA-sequencing), an in vitro selection method, which starts with a random DNA library and yields high-affinity DNA-binding sequences for the studied protein, and ChIP-seq (Chromatin ImmunoPrecipitation DNA-Sequencing).”

Reviewer 3, comment 14: About the length of the CArG-box (e.g. lines 341-342): In many PWMs, the extension seems to occur on only one side, so would a length of 13 bp not be better than 16?

Our response: The CArG-box motif and also the direct flanking sequences constitute a nearly palindromic sequence motif. Whether the final PWM-based motif is 13 or 16 nucleotides long, is in part a consequence of the alignment strategy of the bound and sequenced DNA sequences. It is likely, however, that the MTF dimer recognizes 16 nucleotides since each monomer can make base contacts beyond the CArG-box borders (as discussed in chapter 4.1 Base readout). We prefer to keep 16 bp.

Reviewer 3, comment 15: Line 386: transcription factor function should be MTF function.

Our response: Changed to “MTF”.

Reviewer 3, comment 16: Lines 393 – 395: I would rather say that the I-domain determines PPI-interaction specificity, and thereby the nature of the dimer, and that the nature of the dimer determines DNA-binding specificity through allosteric effects.

Our response: This is equally true. We would like to keep the description in lines 393-395 since this is closer in wording to the original source (Lai et al. 2021).

Reviewer 3, comment 17: Figure 2 is a nice overview that shows all different aspects that contribute to the target site specificity of MTF complexes. The last two aspects (interaction with chromatin modifiers and transcriptional machinery) do not really fit and are not discussed in the text. I would therefore leave these out. Possibly, the authors could add differential binding to additional co-factors (likely via the C-terminus). This most likely also contributes to differential regulation. (instead of showing only target site recognition, it would then be extended to target gene regulation)

Our response: We would like to keep the Figure as it is. The reviewer is right that two aspects are not covered in our manuscript in the same way as the others, but it probably doesn’t do harm to provide a broader perspective here, especially since we refer to a published paper that is cited. The differential binding to co-factors has been covered in a nice recent review already (our ref. [10]) that is complementary to our paper, so we wanted to prevent redundancy. We have, therefore, added a statement to the Legend of Figure 2: “The important aspect of co-factor binding to MTFs is largely neglected here, because it has been covered comprehensively in a recent review already [10].”